Chondrogenic differentiation of adipose-derived mesenchymal stem cells induced by L-ascorbic acid and platelet rich plasma on silk fibroin scaffold

Barlian Anggraini aang@sith.itb.ac.id 1
Judawisastra Hermawan 2
Alfarafisa Nayla M. 1
Wibowo Untung A. 2
Rosadi Imam 3
1 School of Life Sciences and Technology, Institute of Technology Bandung , Bandung , West Java , Indonesia
2 Faculty of Mechanical and Aerospace Engineering, Institute of Technology Bandung , Bandung , West Java , Indonesia
3 Klinik Hayandra , Jakarta , Indonesia
Arany Praveen
Electronic publication date: 2018 Nov 19
Publication date: 2018
Volume: 6
Electronic Location ID: e5809
Received 2018 Mar 9; Accepted 2018 Sep 21
Copyright: ©2018 Barlian et al.
Copyright year: 2018
Copyright holder: Barlian et al.
License: This is an open access article distributed under the terms of the Creative Commons Attribution License, which permits unrestricted use, distribution, reproduction and adaptation in any medium and for any purpose provided that it is properly attributed. For attribution, the original author(s), title, publication source (PeerJ) and either DOI or URL of the article must be cited.
License URL: https://creativecommons.org/licenses/by/4.0/

Keywords: Stem cells, Chondrogenesis, Biomaterial scaffold, Osteoarthritis, Bioactive factors

Funding: The authors received no funding for this work.

==============================
Articular cartilage is an avascular tissue with limited regenerative property. Therefore, a defect or trauma in articular cartilage due to disease or accident can lead to progressive tissue deterioration. Cartilage tissue engineering, by replacing defective cartilage tissue, is a method for repairing such a problem. In this research, three main aspects—cell, biomaterial scaffold, and bioactive factors—that support tissue engineering study were optimized. Adipose-derived mesenchymal stem cells (ADSC) that become cartilage were grown in an optimized growth medium supplemented with either platelet rich plasma (PRP) or L-ascorbic acid (LAA). As the characterization result, the ADSC used in this experiment could be classified as Mesenchymal Stem Cell (MSC) based on multipotency analysis and cell surface marker analysis. The biomaterial scaffold was fabricated from the Bombyx morii cocoon using silk fibroin by salt leaching method and was engineered to form different sizes of pores to provide optimized support for cell adhesion and growth. Biocompatibility and cytotoxicity evaluation was done using MTT assay to optimize silk fibroin concentration and pore size. Characterized ADSC were grown on the optimized scaffold. LAA and PRP were chosen as bioactive factors to induce ADSC differentiation to become chondrocytes. The concentration optimization of LAA and PRP was analyzed by cell proliferation using MTT assay and chondrogenic differentiation by measuring glycosaminoglycan (GAG) using Alcian Blue at 605 nm wavelength. The optimum silk fibroin concentration, pore size, LAA concentration, and PRP concentration were used to grow and differentiate characterized ADSC for 7, 14, and 21 days. The cell morphology on the scaffold was analyzed using a scanning electron microscope (SEM). The result showed that the ADSC could adhere on plastic, express specific cell surface markers (CD73, CD90, and CD105), and could be differentiated into three types of mature cells. The silk fibroin scaffold made from 12% w/v concentration formed a 500 µm pore diameter (SEM analysis), and was shown by MTT assay to be biocompatible and to facilitate cell growth. The optimum concentrations of the bioactive factors LAA and PRP were 50 µg/mL and 10%, respectively. GAG analysis with Alcian Blue staining suggested that PRP induction medium and LAA induction medium on 12% w/v scaffold could effectively promote not only cell adhesion and cell proliferation but also chondrogenic differentiation of ADSC within 21 days of culture. Therefore, this study provides a new approach to articular tissue engineering with a combination of ADSC as cell source, LAA and PRP as bioactive factors, and silk fibroin as a biocompatible and biodegradable scaffold.

Introduction

Unlike other connective tissues, articular cartilage is an avascular tissue which lacks a nervous and lymphatic system ( Zhang, Hu & Athanasiou, 2009). This unique property make it difficult for the cartilage to regenerate tissue after being damaged. Following disease, its structure tends to degrade progressively (Hunziker, 2002). The absence of blood supply may also inhibit the wound healing process (Baugé & Boumédiene, 2015), leading to necrosis (Mason et al., 2000), consequently causing a permanent defect in the area. Some methods have been established to treat patients with cartilage tissue defects, such as microfracture, autologus implantation, autograft, allograft, and joint replacement ( Zhang, Hu & Athanasiou, 2009); however, these methods have some disadvantages and cause side effects.

Tissue engineering is promising for creating new cartilage tissue that is expected to become functional and ready to be implanted to replace damaged cartilage tissue. Four main parameters support the success of articular cartilage tissue engineering: cell type, growth factors or bioactive factors, mechanical stimuli, and scaffold material (Kock, Donkelaar & Ito, 2012).

The cell source for tissue engineering can be obtained from several tissues or organs, one of which is adipose tissue. Adipose-derived mesenchymal stem cells (ADSC) are mesenchymal stem cells (MSC) obtained from perivascular white adipose tissue, including subcutaneous adipose tissue (Kishi, Imanishi & Ohara, 2010). The isolation of ADSC is relatively easy and produces a higher yield of cells compared to other adult stem cell source tissues (Johnstone et al., 2013).

Bioactive factors used in tissue engineering control cell proliferation and differentiation (Brochhausen et al., 2009); In their study, L-ascorbic acid (LAA) and platelet rich plasma (PRP) were used as bioactive factors for chondrogenic differentiation of ADSC. Temu et al. (2010) showed that LAA could induce the differentiation of a ATDC5 cell line to chondrocytes. Moreover, LAA can regulate adult stem cell differentiation to some mesenchymal tissues derivatives, such as adipocytes, osteocytes, myocytes, and chondrocytes (Choi et al., 2008). LAA has a role as co-factor in post-translational modification of collagen molecules (Gessin et al., 1988), which is a component of the extracellular matrix (ECM) of chondrocytes. Previous study had shown that PRP could be a good candidate for a bioactive factor due to its composition, such as transforming growth factor-β1 (TGF-β1), platelet-derived growth factor (PDGF), epidermal growth factor (EGF), insulin-like growth factor-1 (IGF-1), and vascular endhotelial growth factor (VEGF) that are important for cell differentiation and proliferation (Pawitan et al., 2014). Moreover, PRP had been shown to stimulate cell proliferation and matrix biosynthesis of mammalian chondrocytes as well as the expression of a chondrogenic marker in a MSC 3D culture (Akeda et al., 2006; Drengk et al., 2009).

The objective of this research was to induce ADSC differentiation to become chondrocytes which were seeded on silk fibroin scaffolds in a PRP induction medium and a LAA induction medium. Silk fibroin was chosen as the biomaterial of the scaffold because it is biodegradable and biocompatible (Wang et al., 2006).

Materials and Methods

Silk fibroin scaffold fabrication

We used silk fibroin fabricated using a salt leaching method as the scaffolds, as previously characterized by Judawisastra & Wibowo (2017). The silk fibroin was obtained from a Bombyx mori cocoon that was degummed to remove sericin protein which can cause biocompatibility and hypersensitivity problems in vivo ( Altman et al., 2003). The cocoon was cut and immersed in 0.05% Na2CO3 solution for 1 h. The cocoon was washed in deionized water to remove residual Na2CO3 solution and then dried in a fume hood overnight.

Dried silk fibroin was diluted in 8 wt% CaCl2-Formic acid solution at room temperature with constant stirring for 15–30 min. The silk fibroin concentration was optimized by the addition of 6 gr, 8 gr, 10 gr, and 12 gr silk fibroin into 8 wt% CaCl2-Formic acid solution. NaCl with a specific particle size was added into the fibroin solution and homogenized. The NaCl particle size was important because the scaffold pore size was determined by it. The optimization of pore size to support ADSC proliferation was performed using MTT assay for 100, 300, and 500 µm pore size on days 1, 3, 5, 7, and 14. Data were taken in triplicate for each scaffold on each observational day. The ratio of NaCl and fibroin solution was 5:1, and then the mixture was dried in the fume hood overnight. The mixture was immersed in a 70% alcohol solution for ±30 min to induce β-sheet formation (Terada et al., 2016), and then the fibroin was immersed in distilled water for 3 days to remove salt residues. Successfully obtained silk fibroin was stored at −80 °C for 30 min for easier cutting. Before further analysis, th silk fibroin was sterilized using an autoclave for 15–20 min at 121 °C ( Sommer et al., 2016).

Adipose-Derived Stem Cells (ADSC) isolation and culture

The ADSC was obtained from human adipose tissue with ethical approval (No. Reg. 0417060790) from The Health Research Ethics Committee from the Faculty of Medicine, Padjajaran University. The ADSC isolation method followed that of Remelia et al. (2016). Adipose tissue was processed enzymatically using H-Remedy recombinant enzyme which was added to adipose tissue in 10% v/v concentration, and then incubated at 37 °C for 1 h. After 1 h, the growth medium Dulbecco’s Modified Eagle’s Medium (DMEM) low glucose (1 g/L) (Sigma)and L-glutamin (4 mM) were added to the sample to inactivate enzymes. Then, the sample was centrifuged for 5 min with 600g. The supernatant was discarded, and a 10 mL red blood cell lysis buffer was added into the pellet and incubated for 5 min at room temperature. The sample was centrifuged for 10 min at 600g and the supernatant was then discarded. The pellet, which is called stromal vascular fraction (SVF), was cultured at 37 °C, 5% CO2 to increase the number of cells or to facilitate cell proliferation. After the cells’ culture reached 80–90% confluency, they were harvested enzymatically using Trypsin-EDTA (0.25%) and cryopreserved in liquid nitrogen.

ADSC characterization

ADSC characterization included specific cell surface marker analysis and differentiation potency evaluation. The specific cell surface marker refers to the protocol in the Human MSC Analysis Kit (BD Stemflow™). ADSC passage 1 that reached >80% confluency were harvested enzymatically using Trypsin-EDTA (0.25%) (Gibco, Thermo Fisher Scientific, Waltham, MA, USA). Cell concentration at 5  × 106–107 cells/mL was resuspended in 1 ml staining buffer. After resuspension, a 100 µL cells solution in the staining buffer was taken and added into the 5 µL hMSC positive cocktail (CD90FITC, CD105PerCP-Cy5.5, CD73 APC), and hMSC negative cocktail (CD34 PE, CD11b PE, CD19 PE, CD45 PE, HLA-DR PE) antibody or isotype and incubated in the dark for 30 min. Then, the sample was washed twice in the staining buffer and resuspended in 300–500 µL staining buffer. Finally, the sample was analyzed using a flowcytometer.

For the differentiation analysis, ADSC passage 2 that reached >80% confluency was harvested enzymatically in Trypsin-EDTA (0.25%) (Gibco). ADSC were cultured in 24 well-plate (1  × 104 sel/well) in the growth medium DMEM Low Glucose (Gibco) supplemented with 10% FBS (Gibco). After reaching 80% confluency, the growth medium was replaced with a chondrogenic induction medium (MesenCult™ Chondrogenesis Differentiation Kit, Vancouver, Canada), osteogenic induction medium (MesenCult™ Osteogenesis Differentiation Kit), and adipogenic induction medium (MesenCult™ Adipogenic Differentiation Medium (Human)). After 7–14 days incubation (for adipogenesis) and/or longer than 14 days (for chondrogenesis and osteogenesis), the cells were observed using an inverted microscope. The cells were fixed in 4% formaldehyde in saline and stained, using Alcian Blue staining which is specific for glycosaminoglycan, one of the components in chondrocytes extracellular matrix; Alizarin Red staining which is specific for mineralized matrix expression as osteoblast marker; and Oil Red O staining for lipid vacuoles in adipocyte marker. The excess dye stain was washed in PBS. The cell observation was performed by inverted microscope.

Biocompatibility analysis of silk fibroin scaffold

Biocompatibility of the silk fibroin scaffold was analyzed using 3-(4,5-Dimethylthiazol-2-yl)-2,5-diphenyltetrazolium bromidefor (MTT Assay). ADSC (1  × 105 cells/mL) were grown on a sterile scaffold (0.5 cm × 0.5 cm × 1 mm) in 96 well-plate. The cells were maintained in growth medium (supplemented with DMEM) and incubated at 37 °C; 5% CO2. The effect on ADSC at days 1, 3, 5, 7, and 14 after being cultured was evaluated using MTT assay. For the cytotoxicity assay,the growth medium was discarded, and then 10 µL of MTT reagent (5 mg/mL) was added into 100 µL growth medium. The cells were incubated in MTT solution for 4 h at 37 °C in the dark. After that, the MTT solution was removed, and 100 µL dimethyl sulfoxide (DMSO) was added to dilute the formazan crystal that had formed. The absorbance of the solution was read at 570 nm wavelength using a microplate reader (Bio-Rad). The observation was repeated three times.

Scanning Electron Microscope (SEM) analysis

ADSC morphology was analyzed using a scanning electron microscope (SEM) (SU 3500; Hitachi, Krefeld, Germany; Center of Advanced Science ITB) after being seeded on the scaffold. ADSC (106 cells/mL) were grown on the sterile scaffold (0.5 cm × 0.5 cm × 1 mm) in 96 well-plate. The cells were maintained in the growth medium and incubated at 37 °C; 5% CO2. SEM analysis was done on cells cultured for 1 and 21 days. Cells were fixed in 100 µL of 2,5% (v/v) glutaraldehyde in 0.1 M cacodylate buffer (Electron Microscopy Science; Hatfield, PA, USA), and incubated for 24 h at 4 °C. The sample was dehydrated using an ascending alcohol series and dried by the freeze drying method for 3 h. The dried sample was sputtered with gold coating and observed under SEM.

Optimization of LAA and PRP concentration

Cell proliferation in optimization LAA and PRP concentration was analyzed using MTT assay, and the cell differentiation was analyzed using Alcian Blue staining. ADSC (1  × 105 cells/mL) were seeded on 96 well-plate. The concentrations of LAA solution in the induction medium were 25 µg/mL, 50 µg/mL, 100 µg/mL, and 200 µg/L. The concentrations of PRP solution (platelet content approximately 1.086  × 106/µL) in the induction medium with 1% heparin added were 5%, 10%, and 20%. The cell cultures were placed in an incubator at 37 °C; 5% CO2 in the induction medium, and replaced every 2 days. Proliferation analysis was observed on days 1, 3, 5, 7, and 14. In addition, the differentiation potency was analyzed using extracellular matrix staining for sulfated-GAG as a chondrocytes marker. ADSC (104 cells/mL) were grown on 24 well-plate, and the cells were incubated at 37 °C; 5% CO2, and the medium was replaced every 2 days. On days 14 and 21, the cells were prepared for Alcian Blue staining and observed using an inverted microscope to evaluate chondrogenic differentiation. Data were taken in triplicate for each group on every observational day. The intensity of blue colour as the staining result indicated that glycosaminoglycan extracellular matrix had formed. Blue colour intensity was quantified using the software Digimizer (MedCalc, Ostend, Belgium).

Analysis of Glycosaminoglycan (GAG) content

ADSC (106 cells/mL) were grown on a scaffold in LAA induction medium and PRP induction medium. On days 7, 14, and 21, the cell culture was washed in PBS before being fixed using an acetone:methanol (1:1) solution at 4 °C for 3 min. One percent Alcian Blue in 3% acetic acid was added into the cell culture. The cells were incubated for 30 min and the overstaining dye was washed in 3% acetic acid and deionized water. One percent of Sodium dodecyl sulfate (SDS) was added to the cell culture and homogenized using a shaker at 200 rpm for 30 min. The absorbance was read using a microplate reader at 605 nm wavelength. The observation was repeated three times.

Result

ADSC characteristics

Observation using a light microscope showed that cells had a spindle-like morphology, which is flattened. The cells had formed lamellipodium, indicating their complete adherence on the polystyrene flask in the standard culture condition. According to Lotfy et al. (2014), ADSC morphology in their standard culture condition was fibroblast-like and fusiform (spindle-like shape). The population doubling time (PDT) of ADSC was 42.9 h, while that of Griffin et al. (2017) was 39,88  ± 4.4 h. This difference in PDT could be due to the cell passage and genetic variation of donors determined by PDT of ADSC.

The cell surface markers that were used to confirm MSC were cluster of differentiation 90 (CD90), CD73, and CD105 as positive markers, and CD45, CD34, CD11b, CD19, HLA-DR, as negative markers. The singlet and sharp peaks in the graphs show that there was a single population of the cells. The positive and negative markers had the singlet and sharp peak. The ADSC expressed the positive markers CD90 (91.38%), CD73 (96.25%), and CD105 (61.33%). However, the sample and isotype peak of the negative markers overlapped, decreasing the percentage (0.38%) (Fig. 1). The MSC must express those three positive markers exceeding 90% of cell population, while the MSC must express negative markers until below 2% (Dominici et al., 2006). Our result showed that ADSC expressed CD90 (91.38%), CD73 (96.25%), and CD105 (61.33%), and the negative markers were expressed only in a small percentage (0.38%) (Fig. 1).

Figure 1 Flow cytometry result of specific ADSC cell surface markers.

Cell expressed positive markers (A) CD90, (B) CD73, (C) CD 105, and (D) negative marker CD45, CD34, CD11b, CD19, HLA-DR.

The multipotency analysis showed that ADSC complied with one of the MSC criteria, which could be differentiated into three mesenchymal stem cell derivative cells: chondrocyte, osteocyte, and adipocyte (Fig. 2).

Figure 2 Multipotency evaluation of ADSC.

(A) Alcian Blue staining, (B) Alizarin Red and (C) Oil Red O in ADSC culture which was induced with differentiation induction medium, and (D) non-staining group. Black arrow showed positive result from each staining group.

Optimization of silk fibroin scaffold

To promote stronger cell adhesion and faster cell growth, often some modifications are made on scaffolds. To achieve the best support of ADSC growth, we optimized silk fibroin concentration and pore size. Figure 3 depicts that the percentage of cell viability increased gradually within 14 days in different silk fibroin concentrations, indicating that the cells seeded on all scaffolds proliferated actively. On day 14, the scaffold made from 8% w/v, 10% w/v, and 12% w/v had higher percentages of cell viability than the control group, and the 12% w/v silk fibroin scaffold had the highest score.

Figure 3 Growth curve of ADSC on scaffold in different silk fibroin concentration.

Figure 4 shows that all scaffolds supported cell proliferation and maintained cell viability. Apparently, pore size could affect ADSC proliferation. The percentage of cell viability in scaffold with 500 µm pore size was the highest; thus, this scaffold appears to have been the optimal one for supporting ADSC proliferation.

Figure 4 Growth curve of ADSC on scaffolds that have different pore size.

Biocompatibility analysis of silk fibroin scaffold

SEM images were taken to characterize the structure of the silk fibroin scaffold. Based on the SEM analysis presented in Fig. 5, the scaffold made from 12%w/v silk fibroin formed 500 µm pores. The scaffold had an average pore size 536 ± 97 µm and an interconnected porous structure.

Figure 5 SEM image represented morphology of scaffold.

Scaffold was made from 12% w/v silk fibroin and had 500 µm pore size (Judawisastra & Wibowo, 2017). (l—l) represents pore size.

It is known that the surface roughness of a scaffold is an important factor in promoting cell attachment (Acharya, Ghosh & Kundu, 2008). Figures 6A–6B shows that ADSC cells formed a fillipodia structure which might be one of the factors that promoted fast attachment of ADSC cells in this study. The SEM image also shows the cell structure that attached to the scaffold on days 1 and 21 (Figs. 6C–6D). The images indicate that the scaffold could support ADSC proliferation, which is in line with the growth curve (Fig. 7).

Figure 6 Morphology of ADSC on scaffold made from 12% w/v silk fibroin and had 500 µm pore size.

Single cell was on (A) day 1 and (B) day 21. Cell population was on (C) day 1 and (D) day 21. Red arrow shows cytoplasm extension, called filopodia. White arrow and the area marked with yellow stripe line show the area covered by cells.

The growth curve of ADSC determined by the MTT assay on the optimized scaffold showed the desired increase of viable cells between days 1–21 (Fig. 7). The curve shows a lag phase of cell growth from days 1 to 7 and the log phase after day 7.

Figure 7 Growth curve of ADSC on scaffold made from 12% w/v Silk Fibroin and had 500 µm pore size.

SEM images of tissue formation on the scaffold were obtained on day 21 to examine the tissue structure as the cell had reached full density on the scaffolds surface. The SEM result showed the presence of two cell populations on the scaffolds. One formed a monolayer and the other showed aggregates. Based on the scaffolds optimization experiments above related to ADSC cell viability, growth, and proliferation, we found that 12% w/v silk fibroin with 500 µm pore size was the best candidate to promote ADSC cell adhesion and tissue formation (Fig. 8).

Figure 8 ADSC population observed on day 21.

There are two types of cell population in (A) monolayer formation and (B) aggregation.

Optimization of LAA and PRP concentration

To assess the best concentration of bioactive factors to induce cell proliferation in our experiment, we added LAA or PRP at various concentrations to growth medium and examined the cell proliferation by MTT assay independently. Figure 9 shows the growth curve of ADSC in the presence of LAA at various concentrations. The graphs depict that 25 µg/mL, 50 µg/mL, 100 µg/mL, and 200 µg/mL LAA increased the percentage of cell viability from day 1 to 21 of observation. Our data showed that the addition of LAA promoted faster cell proliferation than in cells not treated with LAA, and the best LAA concentration was 50 µg/mL. However, there were no significant differences among various concentrations of LAA. This result indicates that LAA supported ADSC cell proliferation.

Figure 9 Growth curve of ADSC in L-Ascorbic Acid (LAA) supplemented medium in various concentrations.

The quantification using Digimizer software showed the blue color intensity in various LAA concentrations (Fig. 10). The result shows that ADSC in LAA supplemented medium secretes more GAG than the cells grown in standard medium. However, the amount of GAG was not significantly different among groups (p > 0.05), and blue color intensity in the 50 µg/mL treatment group tended to be higher compared to the other groups.

Figure 10 Graph of blue colour intensity xomparison from Alcian Blue staining in various LAA concentrations.

Figure 11 showed ADSC proliferation in various PRP concentrations in the induction medium.The cells grown in the induction medium supplemented with 5%, 10%, and 20% PRP showed an increasing percentage of cell viability from days 1 to 21. The percentage of cell viability from various PRP concentrations were higher than that without PRP (supplemented with FBS). This result indicated that PRP supported ADSC cell proliferation. Among various PRP concentrations, cell viability in 10% PRP was higher than the other concentration on every observational day. From this result, we concluded that 10% PRP in the induction medium was the optimum concentration to promote cell proliferation.

Figure 11 Growth curve of ADSC in various Platelet Rich Plasma (PRP) concentration of medium.

Similar to the LAA experiment, GAG production in cells treated with PRP was assessed with Alcian Blue staining. The Alcian Blue intensity of cells treated with various PRP concentrations was quantified (Fig. 12). As shown in Fig. 12, 10% and 20% PRP concentration caused higher production of GAG than the control group (supplemented with FBS); the blue color intensity in the 5% PRP concentration and the FBS group did not show any siginificant difference (p > 0.05). Furthermore, the blue color intensity in 20% PRP concentration was lower than the blue color intensity in 10% PRP concentration. Based on these results, we conclude that 10% PRP concentration had the best potency in chondrogenic differentiation.

Figure 12 Graph of blue colour intensity comparison from Alcian Blue staining in various PRP concentrations.

Evaluation of ADSC differentiation on scaffold in LAA induction medium and PRP induction medium

Further, we examined the best concentration of LAA and the PRP effect on ADSC differentiation grown on optimized scaffold. To assess cell differentiation, production of GAG on the grown culture was analyzed based on Alcian Blue absorbance at 650 nm (Fig. 13). Both the absorbance value of Alcian Blue stained cells in 50 µg/mL LAA and 10% PRP increased gradually from day 7 to 21. The absorbance value in LAA induction medium or PRP induction medium were higher than that of the control group (supplemented with FBS). Based on the graph, 50 µg/mL of LAA or 10% PRP induction medium could induce chondrogenic GAG production of ADSC grown on scaffold.

Figure 13 Graph of glycosaminoglycan (GAG) content in ADSC cultured on scaffold in 50 µg/mL LAA and 10% PRP supplemented medium.

Discussion

Cartilage tissue damage can be overcome by implanting cartilage tissue in the damaged area, but sources of cartilage tissue are limited. In this research,we utilized MSC from adipose tissue (ADSC) to be differentiated to chondrocytes. Based on multipotency assessment and morphology observation, we confirmed that the ADSC used in this study complied with criteria described for healthy and potent ADSC. Our ADSC showed an ability to differentiate into chondrocytes, adipocytes, and osteocytes. Furthermore, it also complied with the standards of the International Society for Cellular Therapy(ISCT) (Dominici et al., 2006).

Following the international standard for MSC surface markers declared by ISCT, further assessment of ADSC used in this study showed it complied with those criteria, except lower expression of endoglin marker (CD105) than the standard. However, CD105 expression of ADSC has been described as inconsistent and depends on many factors such as cell source, cell passage, isolation method, incubation duration, and growth phase of cells ( Baer & Geiger, 2012). Early passage cells used in our experiment might be the reason for lower expression of CD105 (Crisan et al., 2008). However, based on other markers and morphology analysis, we concluded our grown culture matches ADSC. CD105 is not the main marker to characterize MSC from adipose tissue, and it is recommended only as an alternative or additional marker (Bourin et al., 2013).

The multipotency evaluation of ADSC showed that the cells could be differentiated into chondrocytes, osteocytes, and adipocytes. The same result was obtained by Hamid et al. (2012) showing that ADSC has the differentiation capacity to become chondrocyte, osteocyte, and adipocyte when cultured in a standard induction medium. Therefore, ADSC used in this research complied with MSC criteria.

Based on the optimization of the scaffold, the optimum cell proliferation was obtained on the 12% silk fibroin scaffold. Based on previous study, the mechanical strength of a scaffold is an important factor in articular cartilage tissue engineering because cartilage has a role as facilitator in body mass transmission in movement (Fox, Bedi & Rodeo, 2009). We speculatethat higher silk fibroin concentration increased the mechanical strength of scaffold. The optimum cell proliferation was also facilitated by the scaffold pore size 500 µm, which is the best pore size obtained in a previous experiment (Judawisastra & Wibowo, 2017). The result of our research was in line with Murphy, Haugh & O’Brien (2010) concluding that a scaffold with a bigger pore size facilitates the cells to proliferate faster than the cells seeded on a scaffold with small pore size. Bigger pore size may also facilitate faster cell migration. The pore structure also affects the roughness of the scaffold’s surface, which is one of the important factors for cell attachment (Acharya, Ghosh & Kundu, 2008).

This research showed that within 21 days there were only two phases of growth curve pattern, compared to in the standard culture condition which has three phases in shorter observational days. This pattern was different from the ADSC growth pattern in the standard culture condition. According to (Christodoulou et al., 2013), ADSC was in lag phase on days 1–3 and entered log phase in days 3–8; and after day 8, ADSC started entering the stationary phase. The delayed growth of ADSC on the scaffold could be due to the influence of the scaffold used in this research, including the pore size. Therefore, the role of a scaffold in facilitating ADSC growth needs further investigation. Moreover, on day 21, two types of cell population were found on the scaffold: monolayer and aggregates; however, the existence of those cellpopulation types needs further investigation. According to Zhang et al. (2014), aggregation formation is very important in the initial stage of chondrogenic differentiation because it can facilitate extracellular matrix condensation.

Here, we confirmed that chondrogenesis can be induced by LAA in standard growth medium. Potdar & D’Souza (2010) also showed that an LAA addition in 250 µM concentration into subcutaneous adipose tissue (SCAT) hMSC culture can increase the proliferation rate. The difference in LAA concentration that can increase cell proliferation rate could be due to the difference of cell source donor. Choi et al. (2008) also showed that the role of LAA in MSC proliferation was dose-dependent. Moreover, chondrogenesis evaluation of cells treated with LAA has shown that 50 µg/mL (170 µM) of LAA could induce optimum chondrogenic differentiation. L-ascorbic acid was well known as inducer in chondrogenesis (Choi et al., 2008; Kao et al., 1990), however, the mechanism has not been studied. Based on analysis of proliferation and chondrogenic differentiation potency, the cells in 50 µg/mL LAA showed the best proliferation rate and differentiation potency. Therefore, 50 µg/mL LAA concentration was chosen as the optimum concentration and used for the next investigation in this research. The LAA concentration in 50 µg/mL (approximately 170 µM) is also a general concentration of LAA used in standard chondrogenic induction medium.

In this research, PRP could also induce chondrogenesis because it contains many growth factors that can induce chondrogenesis in ADSC. In agreement with previous study, our study showed that PRP increases the proliferation rate of ADSC; specifically 10% PRP induces the accumulation of chondrogenic GAG (Liao et al., 2015). As well as LAA, ADSC response to PRP was dose dependent with an optimum concentration of 10% to stimulate ADSC proliferation and differentiation Mardani et al. (2013). Spreafico et al. (2009) showed that PRP addition in culture medium was proved to increase the proteoglycan production, although the mechanism is still unknown. The result of our research was supported by Kim et al. (2012) who observed ADSC differentiation on silk fibroin porous scaffold and showed that ADSC could grow and produce chondrogenic GAG. In comparison to Kim et al. (2012), we also evaluated the PRP addition and found that chondrogenic differentiation showed the best when the cells were induced with PRP. Here, we have shown that the engineered silk fibroin scaffold with certain concentrations of LAA and PRP can be applied to maintain the survival of ADSC and stimulate chondrogenesis, thereby providing a new and optimized method for cartilage tissue engineering.

Conclusion

In conclusion, 50 µg/mL of LAA or 10% PRP induction medium can induce optimum chondrogenic differentiation and growth of ADSC on 12% w/v silk fibroin scaffold which had 500 µm pore size. The combination of scaffold and induction medium could faciltate the proliferation and differentiation of ADSC, leading to chondrocytes. In the future, the result of this research could be developed for cartilage tissue engineering.

Supplemental Information

Supplemental Information 1 Raw data exported from the absorbance score resulted from microplate reader Biorad iMark applied for growth curve analyses for Fig. 7

The cells grown on optimized scaffold, which made from 12% w/v Silk Fibroin and had 500 µm pore size on day 1, 3, 5, 7, 14, and 21. Each observational group was quantified triplicate. The growth curves were plotted from the cell viability (y-axis) and observational time (x-axis). The cell viability was obtained from the absorbance value of MTT assay.

Click here for additional data file.

Supplemental Information 2 Raw data exported from the absorbance score resulted from microplate reader Biorad iMark applied for growth curve analyses for Figs. 3 and 4 to determine the optimum scaffold

There are two parameters that observed in determining the optimum scaffold, which are pore size and natural polymers concentration in developing scaffold. The growth curves were plotted from the cell viability (y-axis) and observational time (x-axis). The cell viability was obtained from the absorbance value of MTT assay. Each observational group was quantified triplicate.

Click here for additional data file.

Supplemental Information 3 Raw data represent graph of blue color intensity comparison from alcian blue staining in various (Fig. 10) LAA Concentration and (Fig. 12) PRP Concentration

The blue color intensity was obtained from the absorbance value (y-axis) exported from Biorad iMark. There are 5 groups in LAA optimization process and 4 groups in PRP optimization process. Each group was repeated triplicate.

Click here for additional data file.

Supplemental Information 4 Raw data represent graph of glycosaminoglycan (GAG) content in ADSC cultured on scaffold in 50 µg/mL LAA and 10% PRP supplemented medium (Fig. 13)

The blue color intensity was obtained from the absorbance value (y-axis) exported from Biorad iMark. There are 3 observational groups, which are ADSC on optimum scaffold with optimum LAA supplemented medium, ADSC on optimum scaffold with optimum PRP supplemented medium and ADSC on polystyrene plate. Each group was repeated triplicate.

Click here for additional data file.

Supplemental Information 5 Raw data represents growth curve of ADSC in various L-Ascorbic Acid (LAA) concentration of medium (Fig. 9) and Platelet Rich Plasma (PRP) concentration of medium (Fig. 11)

To optimize the optimum LAA and PRP concentration, the ADSC were grown on LAA supplemented medium and PRP supplemented medium. The growth curves were plotted from the cell viability (y-axis) and observational time (x-axis). The cell viability was obtained from the absorbance value of MTT assay. Each observational group was quantified triplicate.

Click here for additional data file.

This research was facilitated by Klinik Yayasan Hayandra Peduli and HayandraLab. We thank Regina Giovanni for preparing the manuscipt.

Additional Information and Declarations

Competing Interests

Author Contributions

Human Ethics

Data Availability

The authors declare there are no competing interests.

Anggraini Barlian and Hermawan Judawisastra conceived and designed the experiments, analyzed the data, contributed reagents/materials/analysis tools, prepared figures and/or tables, authored or reviewed drafts of the paper, approved the final draft.

Nayla M. Alfarafisa and Untung A. Wibowo conceived and designed the experiments, performed the experiments, analyzed the data, contributed reagents/materials/analysis tools, prepared figures and/or tables, authored or reviewed drafts of the paper, approved the final draft.

Imam Rosadi conceived and designed the experiments, performed the experiments, analyzed the data, contributed reagents/materials/analysis tools, approved the final draft, provide cells and laboratory tools.

The following information was supplied relating to ethical approvals (i.e., approving body and any reference numbers):

The Health Research Ethics Committee from Faculty of Medicine Padjajaran University granted ethical approval to carry out the study within its facilities (Ethical Application Ref: 0417060790).

The following information was supplied regarding data availability:

The raw data are provided in the Supplemental Files.

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
