# Peer review of "Chondrogenic differentiation of adipose-derived mesenchymal stem cells induced by L-ascorbic acid and platelet rich plasma on silk fibroin scaffold"

_PeerJ, doi:10.7717/peerj.5809_

## Round 0.1 · original submission · Major Revisions

Your work has been reviewed by 3 experts who note several issues that need to be addressed. Please note the emphasis on scientific writing and it is strongly suggested you have this work reviewed by a native English speaking colleague or a professional service.

·

Basic reporting

Poor English

Experimental design

Well presented

Validity of the findings

It is novel

Additional comments

The manuscript entitled “Chondrogenic differentiation of adipose-derived mesenchymal stem cells induced by L-ascorbic acid and platelet rich plasma on silk fibroin scaffold” is a poorly written manuscript. It is riddled with syntax errors and spelling mistakes (some of them have been highlighted in the minor changes) which makes it difficult to understand this manuscript. However, this manuscript has potential data which can be published, provided authors correct this manuscript’s English and incorporate following Major and minor changes.
Major questions:
1) All growth curves should be plotted as % viability than just optical density (OD) values.
2) ADSC differentiation to chondrocytes must be validated by gene expression studies for markers of chondrocyte differentiation (like: collagen II, Runx2, and collagen X). If these results are in line with Alcian Blue staining, it will be a solid validation for differentiation.
Minor comments:
Few examples of spelling errors:
The MSC characteristics were plastic adherence, spesific cell surface marker expression using flow cytometry analysis, and multipotency test for spesific mature cell differentiation (adipocytes, chondrocytes, and osteocytes) by extracellular matrix staining- Spelling is wrong and sentence makes no sense.
Example of Syntax error:
Articular cartilage is avascular tissue, which is different from other connective tissues with the absent of nervous and lymphatic system. Please rephrase this statement like- Articular cartilage is avascular tissue, which is different from other connective tissues due to absence of nervous and lymphatic system.
Clarification:
Figure 5- Scaffold was made from 12%w/v silk fibroin and had 500μm pore size (unpublished data)- ???

·

Basic reporting

The manuscript entitled "Chondrogenic differentiation of adipose-derived mesenchymal stem cells induced by L-ascorbic acid and platelet rich plasma on silk fibroin scaffold" reports an improved method using specific concentration of L-ascorbic acid and Platelet rich plasma to stimulate Adipose derived stem cells toward chondrogenic differentiation. I believe the manuscript still requires some minor revision before further consideration for publication.

Experimental design

The experimental design and methods used in this manuscript are enough for publication.

Validity of the findings

Overall, this manuscript demonstrates the ability of Silk fibroin to induce Adipose derived stem cells differentiation in the presence of 50μg/mL of L-ascorbic acid or 10% of Platelet rich plasma induction medium. The methods of assay are sound, the use of in vitro model is adequate, and the reported data appear to make the case that the L-ascorbic acid or Platelet rich plasma play a key role in chondrogenic differentiation.

Additional comments

General Comments for the author:

1. There are grammar errors in the manuscript and the tenses are confused. Therefore, the manuscript should be thoroughly edited.
2. Figure 13 which shows Graph of glycosaminoglycan (GAG) content in ADSC cultured on scaffold is 50μg/mL L-ascorbic acid. Whereas the authors have mentioned only 50µg in the text? Please explain it.
3. Statistical analysis are needed for the Figure 03, 04, 09, 10, 11, 12, and 13. The significant difference in each groups should discuss in the “Result section” and the symbol should be mentioned in each “Figure and Figure caption section”.
4. The authors mentioned pore size “500μm” is Figure 05, 06, and 07. However, there is no data in the manuscript that can show the pore size is 500μm. The authors need to add the data or remove them from the manuscript.
5. In “Figure caption section” more detail description are needed to add for making clear easily follow.
6. I advise the authors to correct the grammatical errors because there are a large number of grammatical mistakes or odd turns of phrase, and too many for a reviewer to list.

·

Basic reporting

Barlian et al show that culturing adipose-derived mesenchymal stem cells on silk fibroin scaffold along with L-ascorbic acid or platelet rich plasma induced chondrogenic differentiation. The multipotency of the cells was confirmed by staining for adipocyte, chondrocyte, and osteocyte markers. The authors analyze the morphology and proliferation of these cells over multiple days.

Although the authors present a relevant study useful in tissue engineering and clinical applications, unfortunately the current manuscript severely lacks novelty, diligence, and quality of the write-up. Owing to numerous grammatical errors, spelling mistakes, incoherent writing, and lack of adequate experimental design, the study is not suitable in the current form for publication. If English is not the native language of the authors, it is highly desirable to get the manuscript corrected by a colleague.

In addition to major concerns regarding grammatical/spelling errors, lack of details about replicates, and non-rigorous interpretation, following are some of the minor concerns:
1. In addition of staining, it would be useful to confirm the differentiation status using gene expression study.
2. In Figure 1, Y-axis is not labelled. It is not mentioned how the gates were decided or what was the control used. In panel C, the 61.33% positivity for CD105 is confusing. Panel C is not labelled properly.
3. Figure 2: It would be useful to show the staining when differentiation condition was not included. Scale bars are missing. The label D is placed on panel C.
4. Figure 3: For all the graphs, inclusion of trendline is distracting. Was the starting number normalized? In figure 3 and 4 some values are high at day 1 itself. It is desirable to normalize the value and present it as %. The significance needs to be calculated.
5. The conclusions made from SEM images are conjectures. It is difficult to agree to the claim that monolayers and aggregates are two distinct populations.
6. Authors need to clearly show that culturing cells on scaffold had an advantage with respect differentiation.
7. The cell number used is inconsistent throughout.
8. Was the pore-size constantly 500um? Was there a range?
9. Some selected issues with grammar/spelling are:
• Absence instead of absent (line 48)
• Lack of ‘the’ at many places
• Line 62-63, 66 needs fixing
• Delete ‘be’ in 71, 72
• *Mammalian (76)
• Many sentences in methods section written in future tense
• *Specific (114, 157)
• *Supernatant (130, 132)
• Inconsistency in space after units
• Inconsistency when writing decimals (comma/periods)
• *Overcome, *Damaged (329)
• *inconsistent (340)
• *Stationary (365)

Experimental design

The experimental design lacks novelty The data is not robust and lacks several controls. The number of replicates is not explicitly mentioned. Conclusions are factual observations and lack inference. The article fails to match the minimum quality of professional English expected. The references are inadequate.

Validity of the findings

Some conclusions require further confirmation of observations using additional experiments such as gene expression analysis. The proliferation assay data is not sufficient to justify the claims made.

---

## Round 0.2 · Minor Revisions

The reviewer has noted this work has been suitably revised as per his comments. However, there remain significant concerns regarding the language editing and presentation. Kindly have the manuscript proof-read by a native English speaking colleague or a professional service to ensure the presented work is properly interpreted.

·

Basic reporting

Article has potential, but still written in a horrible English. It has to be corrected before it is accepted

Experimental design

ok

Validity of the findings

no comments

Additional comments

This manuscript is a potential one. However, it is written in a horrible English. Some of the mistakes have been corrected by the authors but still it needs major revision for its English.
Some of the examples are highlighted below:

Abstract line 25: Biocompatibility and cytoxicity evaluation was done using MTT assay to optimize silk fibroin concentration and pore size.
Cytotoxicity spelling is wrong.
Abstract: Line 59: The result showed that the ADSC could adhere on plastic, express spesific cell surface markers (CD73, CD90, dan CD105), and be differentiated into three types of mature cells.
Specific spelling is wrong.
Abstract: line 60-63: As the result, silk fibroin scaffold made from 12% w/v concentration formed 500µm pore diameter (SEM analysis), is shown by MTT assay to be biocompatible and facilitate cell growth.
It should be: As a result.

---

## Round 0.3 · accepted · Accept

Please correct the spelling error on Line 33: 'and' instead of 'dan'

#